# Optimally Deceiving a Learning Leader in Stackelberg Games

**Georgios Birmpas**
Sapienza University of Rome
gebirbas@gmail.com

**Jiarui Gan**
University of Oxford
jiarui.gan@cs.ox.ac.uk

**Alexandros Hollender**
University of Oxford
alexandros.hollender@cs.ox.ac.uk

**Francisco J. Marmolejo-Cossío**
University of Oxford
francisco.marmolejo@cs.ox.ac.uk

**Ninad Rajgopal**
University of Oxford
ninad.rajgopal@cs.ox.ac.uk

**Alexandros A. Voudouris**
University of Essex
alexandros.voudouris@essex.ac.uk

## Abstract

Recent results in the ML community have revealed that learning algorithms used to compute the optimal strategy for the leader to commit to in a Stackelberg game, are susceptible to manipulation by the follower. Such a learning algorithm operates by querying the best responses or the payoffs of the follower, who consequently can deceive the algorithm by responding as if their payoffs were much different than what they actually are. For this strategic behavior to be successful, the main challenge faced by the follower is to pinpoint the payoffs that would make the learning algorithm compute a commitment so that best responding to it maximizes the follower's utility, according to the true payoffs. While this problem has been considered before, the related literature only focused on the simplified scenario in which the payoff space is finite, thus leaving the general version of the problem unanswered. In this paper, we fill this gap by showing that it is always possible for the follower to *efficiently* compute (near-)optimal payoffs for various scenarios of learning interaction between the leader and the follower.

## 1 Introduction

*Stackelberg games* are a simple yet powerful model for sequential interaction among strategic agents. In such games there are two players: a leader and a follower. The leader commits to an action, and the follower acts upon observing the leader's commitment. The simple sequential structure of the game permits modeling a multitude of important scenarios. Indicative applications include the competition between a large and a small firm [43], the allocation of defensive resources [42], and the competition among mining pools in the Bitcoin network [32, 41].

In Stackelberg games, the leader is interested in finding the best commitment she can make, assuming that the follower behaves rationally. The combination of such a commitment by the leader and the follower's rational best response to it leads to a strong Stackelberg equilibrium (SSE). In general, the utility that the leader obtains in an SSE is larger than what she would obtain in a Nash equilibrium of the corresponding one-shot game [40], implying that the leader prefers to commit than to engage in a simultaneous game with the follower.

In case the leader has access to both hers and the follower's payoff parameters, computing an SSE is a computationally tractable problem [15]. In practice however, the leader may have limited or no information about the follower's payoffs. Consequently, in order to determine the optimal commitment, the leader must endeavor to elicit information about the incentives of the follower through indirect means. This avenue of research has led to a plethora of active-learning-based approaches for the computation of SSEs [3, 6, 30, 36, 39]. At the same time, inspired by recent developments in the ML community regarding adversarial examples in classification algorithms [4, 31], there has been a stream of recent papers exploring the notion of adversarial deception by the follower, when facing algorithms used by the leader for learning SSEs in Stackelberg games.

Specifically, when an algorithm learns an SSE by querying the follower's best responses, the follower can use fake best responses to distort the SSE learned by the algorithm. As recently explored by Gan *et al.* [20], one particular approach the follower can employ, is to imitate best responses implied by payoffs that are different from his actual ones. The key to the success of such a deceptive behavior is thus to pinpoint the fake payoffs that could make the leader learn an SSE in which the actual utility of the follower is maximized. In the scenario studied in [20], this task is trivial as the follower's choices are limited to a finite set that has a size bounded by the size of the problem representation; thus, to efficiently find out the optimal payoffs, the follower can simply enumerate all possible matrices.

To the best of our knowledge, the general version of this problem, where the follower is allowed to use *any* payoff matrix, without restrictions on the space of possible values, has been considered only in two very recent papers [19, 34], which however focused on the specific application of Stackelberg games to security resource allocation problems. Besides that, no progress has been made for general Stackelberg games. In this paper, we aim to fill in this gap, by completely resolving this computational problem, a result that reflects the insecurity of learning to commit in Stackelberg games.

**Our Contribution.** We explore how a follower can optimally deceive a learning leader in Stackelberg games by misreporting his payoff matrix, and study the tractability of the corresponding optimization problem. As in previous work, our objective is to compute the fake payoff matrix according to which the follower can best respond to make the leader learn an SSE in which the true utility of the follower is maximized. However, unlike the related literature, we do not impose any restrictions on the space from which the payoffs are selected or on the type of the game.

By exploiting an intuitive characterization of all strategy profiles that can be induced as SSEs in Stackelberg games, we show that it is always possible (irrespective of the learning algorithm employed by the leader) for the follower to compute, in polynomial time, a payoff matrix implying an SSE which maximizes his true utility. Furthermore, we strengthen this result to resolve possible equilibrium selection issues, by showing that the follower can construct a payoff matrix that induces a *unique* SSE, in which his utility is maximized up to some arbitrarily small loss.

**Other Related Work.** Our paper is related to an emerging line of work at the intersection of machine learning and algorithmic game theory, dealing with scenarios where the samples used for training learning algorithms are controlled by strategic agents, who aim to optimize their personal benefit. Indicatively, there has been recent interest in the analysis of the effect of strategic behavior on the efficiency of existing algorithms, as well as the design of algorithms resilient to strategic manipulation for linear regression [5, 13, 16, 28, 37, 44] and classification [12, 17, 33, 45].

Beyond the strategic considerations above, our work is also related to the study of query protocols for learning game-theoretic equilibria. In this setting, as in ours, algorithms for computing equilibria via utility and best response queries are a natural starting point. For utility queries, there has been much work in proving exponential lower bounds for randomized computation of exact, approximate and well-supported Nash equilibria [2, 1, 11, 24, 26, 27], as well as providing query-efficient protocols for approximate Nash equilibrium computation in bimatrix games, congestion games [18], anonymous games [25], and large games [23]. Best response queries are weaker than utility queries, but they arise naturally in practice, and are also expressive enough to implement fictitious play, a dynamic first proposed in [10], and proven to converge in [38] for two-player zero-sum games to an approximate Nash equilibrium. In terms of equilibrium computation, the authors in [22] also provide query-efficient algorithms for computing approximate Nash equilibria for bimatrix games via best response queries provided one agent has a constant number of strategies.

Finally, learning via incentive queries in games is directly related to the theory of preference elicitation, where the goal is to mine information about the private parameters of the agents by interacting with them [7, 29, 46, 21]. This has many applications, most notably combinatorial auctions, where access to the valuation functions of the agents is achieved via value or demand queries [8, 14, 35].

## 2  Preliminaries

A Stackelberg game (SG) is a sequential game between a *leader* and a *follower*; we refer to the leader as a female and to the follower as a male. The leader commits to a strategy, and the follower then acts upon observing this commitment. We consider finite SGs, in which the leader and the follower have $m$ and $n$ *pure strategies* at their disposal, respectively, and their utilities for all possible outcomes are given by the matrices $u^L, u^F \in \mathbb{R}^{m \times n}$. The entries $u^L(i,j)$ and $u^F(i,j)$ denote the utilities of the leader and the follower, under *pure strategy profile* $(i,j) \in [m] \times [n]$. We use $\mathcal{G} = (u^L, u^F)$ to denote the SG with payoff matrices $u^L$ and $u^F$; we omit $m$ and $n$ as they are clear from context.

Like one-shot games, the agents are allowed to employ mixed strategies whereby they randomize over actions in their strategy set. A mixed strategy of the leader is a probability distribution over $[m]$, denoted by $\mathbf{x} \in \Delta^{m-1} = \{\mathbf{x} \geq 0 : \sum_{i \in [m]} x_i = 1\}$. By slightly abusing notation, we let $u^L(\mathbf{x}, j) = \sum_{i \in [m]} x_i \cdot u^L(i,j)$ be the *expected utility* of the leader when she plays the mixed strategy $\mathbf{x}$ and the follower plays a pure strategy $j$. Similarly, we define $u^F(\mathbf{x}, j) = \sum_{i \in [m]} x_i \cdot u^F(i,j)$ for the follower. For a given mixed strategy $\mathbf{x} \in \Delta^{m-1}$ of the leader, we say that $j \in [n]$ is a *follower best response* if $u^F(\mathbf{x}, j) = \max_{\ell \in [n]} u^F(\mathbf{x}, \ell)$; we denote the set of all follower best responses to $\mathbf{x}$ by $BR(\mathbf{x}) \subseteq [n]$ and refer to the function $BR$ as the *best response correspondence* of the follower.

A *strong Stackelberg equilibrium* (SSE) is the standard solution concept in SGs, and captures the situation where the leader commits to a mixed strategy that maximizes her expected utility, while taking into account the follower's best response to her commitment. It is assumed that the follower breaks ties in favor of the leader when he has multiple best responses.[1]

**Definition 2.1** (SSE). A strategy profile $(\mathbf{x}, j)$ is an SSE of the SG $\mathcal{G} = (u^L, u^F)$ if

$$(\mathbf{x}, j) \in \arg\max_{\mathbf{y} \in \Delta^{m-1}, \ell \in BR(\mathbf{y})} u^L(\mathbf{y}, \ell).$$

**Learning SSEs and Deceptive Follower Behavior.**   We consider the scenario where the leader has full knowledge of her utility matrix $u^L$, and aims to compute an SSE by interacting with the follower and gleaning information about $u^F$. For example, the leader could observe follower best responses in play (akin to having query access to $BR$), or observe follower payoffs at pure strategy profiles during play (akin to having query access to $u^F$ as a function). Hence, this can be cast as the problem of learning an SSE with a specified notion of query access to information about the follower's incentives.[2]

Consider an SG $\mathcal{G} = (u^L, u^F)$. If the follower controls the flow of information to the leader in this paradigm, he may consider perpetually interacting with the leader as if he had a different payoff matrix $\tilde{u}^F$, which can make the leader believe that both agents are playing the game $\widetilde{\mathcal{G}} = (u^L, \tilde{u}^F)$. This deceiving power provides the follower with an incentive to act according to $\widetilde{\mathcal{G}}$ for a judicious choice of $\tilde{u}^F$, because the SSEs in $\widetilde{\mathcal{G}}$ may provide larger utility (according to $u^F$) than the SSEs in $\mathcal{G}$. More concretely, the example below shows that the follower can gain an arbitrary benefit by deceiving the leader to play a different game. Moreover, the example also shows that the leader's utility loss can be arbitrarily bad.

**Example 2.2** (**Beneficial deception**). Let $\alpha \in [0, 1]$ and consider the following matrices:

$$R = \begin{pmatrix} 1 & 0 \\ 0 & 0 \end{pmatrix}, \quad C_\alpha = \begin{pmatrix} 0 & \alpha \\ 1 & \alpha \end{pmatrix}$$

Now, suppose that $u^L = R$ and $u^F = C_\alpha$, and let $x \in [0, 1]$ represent the probability mass that the leader (row player) places on the first row (her first strategy); thus, $1 - x$ is the probability with which she plays her second strategy. Given this mixed strategy of the leader, the utilities that the follower expects to derive from her two strategies (columns) are $u^F(x, 1) = 1 - x$ and $u^F(x, 2) = \alpha$. Consequently, the first strategy is a best response of the follower when $x \in [0, 1 - \alpha]$, and the second one is a best response when $x \in (1 - \alpha, 1]$ (when $x = 1 - \alpha$, the tie is broken in favor of the leader). With this information, it is clear that the SSE of the game occurs when the leader chooses $x = 1 - \alpha$ and the follower plays his first strategy. As a result, the follower's utility is $u^F(1 - \alpha, 1) = \alpha$.

However, for any $\alpha < 1$, the follower has an incentive to deceive the leader into playing the game $\widetilde{\mathcal{G}} = (R, C_1)$, which will improve his utility in the resulting SSE to 1. This will be an improvement by a multiplicative factor of $1/\alpha$, which can be arbitrarily large when $\alpha$ is arbitrarily close to 0. □

**Inducible Strategy Profiles.** The ultimate goal of the follower is to identify the SSE that maximizes his true utility, from the set of SSEs that he can deceive the leader into learning. We will refer to such SSEs as *inducible strategy profiles*. At a high level, the follower's problem can now be expressed as the following optimization problem:

$$\max_{\mathbf{x}, j} \quad u^F(\mathbf{x}, j), \quad \text{subject to} \quad (\mathbf{x}, j) \text{ is inducible} \tag{1}$$

This maximum utility for the follower is called the *optimal inducible utility*. If the maximum value is never achieved, then for every $\varepsilon > 0$, we would like to be able to find an inducible SSE that achieves a value $\varepsilon$-close to the supremum value.

As discussed previously, the leader can learn an SSE by gleaning information about the incentives of the follower by querying the best responses of the follower to particular leader strategies, or more refined information about the follower's payoff matrix. Depending on the type of information queried, we can define various notions of inducible strategy profiles.

In more detail, suppose the leader can only query the best responses of the follower, who behaves according to some best response correspondence $\widetilde{BR} : \Delta^{m-1} \to 2^{[n]} \setminus \{\varnothing\}$. This interaction between the leader and the follower leads to a game $\widetilde{\mathcal{G}} = (u^L, \widetilde{BR})$ where only information about $\widetilde{BR}$ is known (instead of a payoff matrix implying $\widetilde{BR}$). The definition of $\widetilde{BR}$ enforces a best response answer to any possible query. Consequently, the leader learns an SSE $(\mathbf{x}, j) \in \arg\max_{\mathbf{y} \in \Delta^{m-1}, \ell \in \widetilde{BR}(\mathbf{y})} u^L(\mathbf{y}, \ell)$, which yields the following notion of *BR-inducible* strategy profiles.

**Definition 2.3** (BR-inducibility). A strategy profile $(\mathbf{x}, j)$ is *BR-inducible* with respect to $u^L$ if there exists a best response correspondence $\widetilde{BR} : \Delta^{m-1} \to 2^{[n]} \setminus \{\varnothing\}$ such that $(\mathbf{x}, j)$ is an SSE of the game $\widetilde{\mathcal{G}} = (u^L, \widetilde{BR})$, in which case we say that $(\mathbf{x}, j)$ is *induced* by $\widetilde{BR}$.

Next, consider the case where the leader can query information about the payoffs of the follower, who can now behave according to a fake payoff matrix $\tilde{u}^F$. We refer to the SSEs of the resulting game $\widetilde{\mathcal{G}} = (u^L, \tilde{u}^F)$ as *payoff-inducible* strategy profiles.

**Definition 2.4** (Payoff-inducibility). A strategy profile $(\mathbf{x}, j)$ is said to be *payoff-inducible* with respect to $u^L$ if there exists $\tilde{u}^F \in \mathbb{R}^{m \times n}$ such that $(\mathbf{x}, j)$ is an SSE in the game $\widetilde{\mathcal{G}} = (u^L, \tilde{u}^F)$, in which case we say that $(\mathbf{x}, j)$ is *induced* by $\tilde{u}^F$.

Clearly, payoff-inducibility is stricter than BR-inducibility: for every choice of $\tilde{u}^F$, the corresponding best response correspondence $\widetilde{BR}(\mathbf{y}) = \arg\max_{\ell \in [n]} \tilde{u}^F(\mathbf{y}, \ell)$ induces the same SSEs as $\tilde{u}^F$ does.

Note that the above definitions only require an inducible strategy profile to be a verifiable SSE, with respect to the information about the follower's incentives (either $\widetilde{BR}$ or $\tilde{u}^F$). However, it may happen that the resulting game $\widetilde{\mathcal{G}}$ has multiple SSEs, which gives rise to an equilibrium selection issue. Indeed, in practice, it is not realistic to assume that the follower has any control over which SSE is chosen by the leader (who moves first in the game). To address this, and thus completely resolve the optimal deception problem for the follower, we introduce an even stricter notion of inducibility on top of payoff-inducibility, which requires $\widetilde{\mathcal{G}}$ to have a unique SSE.

**Definition 2.5** (Strong inducibility). A strategy profile $(\mathbf{x}, j)$ is said to be *strongly inducible* with respect to $u^L$, if there exists a matrix $\tilde{u}^F \in \mathbb{R}^{m \times n}$ such that $(\mathbf{x}, j)$ is the *unique* SSE of the game $\widetilde{\mathcal{G}} = (u^L, \tilde{u}^F)$, in which case we say that $(\mathbf{x}, j)$ is *strongly* induced by $\tilde{u}^F$.

In the next sections, we will investigate solutions to (1) under the inducibility notions above, from the weakest to the strongest. Our general approach is to decompose (1) into $n$ sub-problems by enumerating all possible follower responses $j \in [n]$. For each strategy $j$, we solve the corresponding optimization problem, and pick the one that yields the maximum utility for the follower. Due to space constraints, some proofs are omitted and can be found in the supplementary material.

## 3 Best Response Inducibility

Let us start our analysis by considering the case in which the leader queries the best responses of the follower. The aim of the follower is to deceive the leader towards a strategy profile that is BR-inducible; see Definition 2.3. Indeed, if the follower is allowed to use an arbitrary $\widetilde{BR}$ to induce a strategy profile $(\mathbf{x}, j)$, he can simply define $\widetilde{BR}$ as follows:

$$\widetilde{BR}(\mathbf{y}) = \begin{cases} \{j\} & \text{if } \mathbf{y} = \mathbf{x} \\ \arg\min_{\ell \in [n]} u^L(\mathbf{y}, \ell) & \text{if } \mathbf{y} \neq \mathbf{x}. \end{cases}$$

Namely, the follower threatens to choose the worst possible response against any leader strategy $\mathbf{y} \neq \mathbf{x}$, so as to minimize the leader's incentive to commit to these strategies. This $\widetilde{BR}$ will successfully convince the leader that $(\mathbf{x}, j)$ is an SSE of $\widetilde{\mathcal{G}}$, hence inducing $(\mathbf{x}, j)$, if the threat is powerful enough, that is, if $u^L(\mathbf{x}, j) \geq \min_{\ell \in [n]} u^L(\mathbf{y}, \ell)$ for all $\mathbf{y} \in \Delta^{m-1}$. Equivalently, this means that

$$u^L(\mathbf{x}, j) \geq M := \max_{\mathbf{y} \in \Delta^{m-1}} \min_{\ell \in [n]} u^L(\mathbf{y}, \ell), \tag{2}$$

where $M$ is exactly the leader's *maximin utility*. Indeed, (2) is necessary for $(\mathbf{x}, j)$ to be BR-inducible: if on the contrary $u^L(\mathbf{x}, j) < M$, then by committing to $\mathbf{y}^* \in \arg\max_{\mathbf{y} \in \Delta^{m-1}} \min_{\ell \in [n]} u^L(\mathbf{y}, \ell)$, the leader can obtain (at least) her maximin utility, which will be strictly larger than $u^L(\mathbf{x}, j)$.

Thus, condition (2) gives a simple criterion for BR-inducibility. The problem is that such $\widetilde{BR}$ may be far from being one that arises from a choice of $\tilde{u}^F$. To alleviate this limitation, we impose a stricter condition on $\widetilde{BR}$, which can also be viewed as an intermediate step towards our goal of studying payoff inducibility.

**Polytopal BR Correspondence.** In a similar vein to [22], we require that, for every $\ell \in [n]$, the set of leader strategies to which $\ell$ is a best response $\widetilde{BR}^{-1}(\ell) = \{\mathbf{y} \in \Delta^{m-1} : \ell \in \widetilde{BR}(\mathbf{y})\}$ is a closed convex polytope, and the union of all these sets forms a partition of $\Delta^{m-1}$ (for example, see the polytope partition of $\Delta^2$ in Figure 1). Any best response correspondence $\widetilde{BR}$ satisfying this assumption is called *polytopal* (see the supplementary material for a formal definition). Being polytopal is necessary for $\widetilde{BR}$ to arise from some payoff matrix. Indeed, it can be shown that the *true* best response correspondence $BR$ that arises from $u^F$ is polytopal. Thus, if the follower attempts to deceive the leader via a fake $\widetilde{BR}$, the leader might spot the deception in case $\widetilde{BR}$ is not polytopal.

It turns out that the following correspondence, which we denote as $\widetilde{BR}_{\mathrm{P}}$, is polytopal and, as we will shortly show, it is in fact as powerful as any best response correspondence.

$$\widetilde{BR}_{\mathrm{P}}(\mathbf{y}) = \begin{cases} \{j\} & \text{if } \mathbf{y} \in \Delta^{m-1} \setminus \overline{U_j(\mathbf{x})} \\ \{j\} \cup \arg\min_{\ell \in [n] \setminus \{j\}} u^L(\mathbf{y}, \ell) & \text{if } \mathbf{y} \in \overline{U_j(\mathbf{x})} \setminus U_j(\mathbf{x}) \\ \arg\min_{\ell \in [n] \setminus \{j\}} u^L(\mathbf{y}, \ell) & \text{if } \mathbf{y} \in U_j(\mathbf{x}) \end{cases}$$

where $\overline{U_j(\mathbf{x})}$ is the closure of $U_j(\mathbf{x}) = \{\mathbf{y} \in \Delta^{m-1} : u^L(\mathbf{y}, j) > u^L(\mathbf{x}, j)\}$, and $\mathbf{x}$ is the leader's strategy that we want to induce.[3] Intuitively, it is safe for the follower to respond by playing $j$ against any leader strategy $\mathbf{y}$ if $u^L(\mathbf{y}, j) \leq u^L(\mathbf{x}, j)$, in which case the leader does not have a strong incentive to commit to $\mathbf{y}$ instead of $\mathbf{x}$. In response to the other strategies, however, the follower

$$u^L = \begin{pmatrix} 0 & 1 & 1 \\ 1 & -1/2 & 1/2 \\ 1 & 1/2 & -1/2 \end{pmatrix}$$

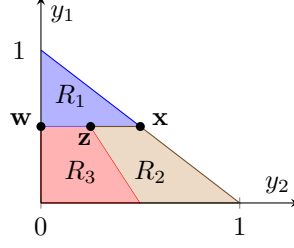

Figure 1: No payoff matrix $\tilde{u}^F$ realizes the polytopal BR correspondence $\widetilde{BR}_{\mathrm{P}}$, such that $\ell \in \widetilde{BR}_{\mathrm{P}}$ if and only if $\mathbf{y} \in R_\ell$, where $R_1 = \{\mathbf{y} \in \Delta^2 : y_1 \geq y_2 + y_3\}$, $R_2 = \{\mathbf{y} \in \Delta^2 : y_1 \leq y_2 + y_3 \text{ and } y_2 \geq y_3\}$, and $R_3 = \{\mathbf{y} \in \Delta^2 : y_1 \leq y_2 + y_3 \text{ and } y_2 \leq y_3\}$.

needs to play a different strategy in order to minimize the leader's incentive to commit to such a $\mathbf{y}$. Therefore, this approach will successfully induce $(\mathbf{x}, j)$ if and only if the following holds:

$$u^L(\mathbf{x}, j) \geq \max_{\mathbf{y} \in \overline{U_j(\mathbf{x})}} \min_{\ell \in [n] \setminus \{j\}} u^L(\mathbf{y}, \ell), \tag{3}$$

where we use the convention that $\max \varnothing = -\infty$. It is easy to see that $\widetilde{BR}_{\mathrm{P}}$ is indeed polytopal: $\widetilde{BR}_{\mathrm{P}}^{-1}(j) = \Delta^{m-1} \setminus U_j(\mathbf{x})$ is a closed convex polytope, and the same holds for the sets $\widetilde{BR}_{\mathrm{P}}^{-1}(\ell)$ defined by the hyperplanes $u^L(\mathbf{y}, \ell) \leq u^L(\mathbf{y}, k)$, $k \in [n] \setminus \{j\}$ and the borders of $\overline{U_j(\mathbf{x})}$, which further form a partition of $\overline{U_j(\mathbf{x})}$.

In fact, (2) is equivalent to (3), meaning that the extra condition imposed on $\widetilde{BR}_{\mathrm{P}}$ does not compromise its power: if $(\mathbf{x}, j)$ can be induced by an arbitrary $\widetilde{BR}$ then it can also be induced by $\widetilde{BR}_{\mathrm{P}}$. We state this result in Lemma 3.1.

**Lemma 3.1.** $u^L(\mathbf{x}, j) \geq M$ if and only if $u^L(\mathbf{x}, j) \geq \max_{\mathbf{y} \in \overline{U_j(\mathbf{x})}} \min_{\ell \in [n] \setminus \{j\}} u^L(\mathbf{y}, \ell)$.

Using Lemma 3.1, we can efficiently compute the best strategy profile that can be induced by $\widetilde{BR}_{\mathrm{P}}$, simply by solving the following Linear Program (LP) for each $j \in [n]$:

$$\max_{\mathbf{x} \in \Delta^{m-1}} u^F(\mathbf{x}, j), \quad \text{subject to} \quad u^L(\mathbf{x}, j) \geq M \tag{4}$$

At this point, it might be tempting to think that with the polytopal constraint imposed, we would also be able to construct an explicit payoff matrix $\tilde{u}^F$ to implement $\widetilde{BR}_{\mathrm{P}}$. Unfortunately, this is not true as Example 3.2 illustrates. Surprisingly though, in the next section we will show that, even though we cannot construct a payoff matrix that implements $\widetilde{BR}_{\mathrm{P}}$ directly, every strategy profile $(\mathbf{x}, j)$ that is $\widetilde{BR}_{\mathrm{P}}$-inducible is in fact payoff-inducible. We also present an efficient algorithm for computing a payoff matrix $\tilde{u}^F$ to induce such $(\mathbf{x}, j)$.

**Example 3.2.** Consider a $3 \times 3$ game with the leader payoff matrix given in Figure 1. Let $\widetilde{BR}_{\mathrm{P}}$ be a polytopal BR correspondence defined by the regions $R_1$, $R_2$, and $R_3$ in Figure 1, such that $\ell \in \widetilde{BR}_{\mathrm{P}}$ if and only if $\mathbf{y} \in R_\ell$. This best response behavior cannot be realized by any payoff matrix. To see this, suppose $\widetilde{BR}_{\mathrm{P}}$ is realized by some $\tilde{u}^F \in \mathbb{R}^{3 \times 3}$. Let $\mathbf{x} = (\frac{1}{2}, \frac{1}{2}, 0)$, $\mathbf{w} = (\frac{1}{2}, 0, \frac{1}{2})$, and $\mathbf{z} = (\frac{1}{2}, \frac{1}{4}, \frac{1}{4})$. We have $\widetilde{BR}_{\mathrm{P}}(\mathbf{z}) = \{1, 2, 3\}$ and $\widetilde{BR}_{\mathrm{P}}(\mathbf{w}) = \{1, 3\}$. This means that $u^L(\mathbf{z}, 1) = u^L(\mathbf{z}, 3) = u^L(\mathbf{z}, 2)$ and $u^L(\mathbf{w}, 1) = u^L(\mathbf{w}, 3) > u^L(\mathbf{w}, 2)$. Since $\mathbf{x} = 2\mathbf{z} - \mathbf{w}$, by the linearity of the utility function, $u^L(\mathbf{x}, 1) = u^L(\mathbf{x}, 3) < u^L(\mathbf{x}, 2)$, which contradicts the fact that $\widetilde{BR}_{\mathrm{P}}(\mathbf{x}) = \{1, 2\}$. $\square$

## 4 Payoff Inducibility

In this section, we will show that every profile strategy that can be induced by $\widetilde{BR}_{\mathrm{P}}$ is also payoff-inducible, and a corresponding payoff matrix can be efficiently constructed. Recall that $M = \max_{\mathbf{y} \in \Delta^{m-1}} \min_{\ell \in [n]} u^L(\mathbf{y}, \ell)$ is the maximin utility of the leader. We will show the following characterization as one of our key results, which enables us to use the LP in (4) to efficiently compute a payoff matrix that achieves the optimal inducible utility.

**Theorem 4.1.** *A strategy profile $(\mathbf{x}, j)$ is payoff-inducible if and only if $u^L(\mathbf{x}, j) \geq M$. Furthermore, a matrix $\tilde{u}^F$ inducing $(\mathbf{x}, j)$ can be constructed in polynomial time.*

One direction of the characterization is easy to show. Indeed, if $(\mathbf{x}, j)$ is payoff-inducible, then it is also BR-inducible, and as seen in Section 3, it holds that $u^L(\mathbf{x}, j) \geq M$.

Now consider any profile $(\mathbf{x}, j)$ such that $u^L(\mathbf{x}, j) \geq M$. Recall that $U_j(\mathbf{x}) = \{\mathbf{y} \in \Delta^{m-1} : u^L(\mathbf{y}, j) > u^L(\mathbf{x}, j)\}$. Without loss of generality, in what follows, we can also assume that $U_j(\mathbf{x}) \neq \varnothing$: if $U_j(\mathbf{x}) = \varnothing$, then $(\mathbf{x}, j)$ will be an SSE if the follower always responds by playing $j$; this can easily be achieved by claiming that $j$ strictly dominates all other strategies, i.e., by letting $\tilde{u}^F(i, j) = 1$ and $\tilde{u}^F(i, \ell) = 0$ for all $i \in [m]$ and $\ell \in [n] \setminus \{j\}$.

We begin by analyzing the following payoff function that forms the basis of our approach. Let $\widehat{S} \subseteq [n] \setminus \{j\}$ and pick $k \in \operatorname{argmin}_{\ell \in \widehat{S}} u^L(\mathbf{x}, \ell)$ arbitrarily. For all $\mathbf{y} \in \Delta^{m-1}$, let

$$\tilde{u}^F(\mathbf{y}, \ell) = \begin{cases} -u^L(\mathbf{y}, \ell) & \text{if } \ell \in \widehat{S} \\ -u^L(\mathbf{y}, k) - 1 & \text{if } \ell \in [n] \setminus (\widehat{S} \cup \{j\}) \\ -u^L(\mathbf{y}, k) + \alpha\left(u^L(\mathbf{x}, j) - u^L(\mathbf{y}, j)\right) & \text{if } \ell = j \end{cases} \tag{5}$$

where $\alpha > 0$ is a constant. In what follows, we will let $\widetilde{BR}$ denote the best response correspondence corresponding to $\tilde{u}^F$, i.e., $\widetilde{BR}(\mathbf{y}) = \operatorname{argmax}_{\ell \in [n]} \tilde{u}^F(\mathbf{y}, \ell)$. Note that we can compute the payoff matrix corresponding to $\tilde{u}^F$ in polynomial time. Then, the hope is that with appropriately chosen $\widehat{S}$ and $\alpha$, the payoff matrix will induce $(\mathbf{x}, j)$. Indeed, $\tilde{u}^F$ has the following nice properties:

    i. Strategy $j$ is indeed a best response to $\mathbf{x}$, since, by the choice of $k$ we have $\tilde{u}^F(\mathbf{x}, j) = -u^L(\mathbf{x}, k) \geq -\min_{\ell \in \widehat{S}} u^L(\mathbf{x}, \ell) = \max_{\ell \in \widehat{S}} \tilde{u}^F(\mathbf{x}, \ell)$.

    ii. Any $\ell \in [n] \setminus (\widehat{S} \cup \{j\})$ cannot be a best response of the follower as it is strictly dominated by $k$, i.e., $\tilde{u}^F(\mathbf{y}, \ell) < \tilde{u}^F(\mathbf{y}, k)$ for all $\mathbf{y} \in \Delta^{m-1}$. Thus, $\widetilde{BR}(\mathbf{y}) \subseteq \widehat{S} \cup \{j\}$ for all $\mathbf{y} \in \Delta^{m-1}$.

    iii. If $j$ is a best response to some $\mathbf{y} \in \Delta^{m-1}$, then $u^L(\mathbf{y}, j) \leq u^L(\mathbf{x}, j)$. Indeed, $j \in \widetilde{BR}(\mathbf{y})$ implies that $\tilde{u}^F(\mathbf{y}, j) = \max_{\ell \in [n]} \tilde{u}^F(\mathbf{y}, \ell) \geq \tilde{u}^F(\mathbf{y}, k)$. Substituting $\tilde{u}^F(\mathbf{y}, j) = -u^L(\mathbf{y}, k) + \alpha\left(u^L(\mathbf{x}, j) - u^L(\mathbf{y}, j)\right)$ into this inequality and rearranging the terms immediately gives $u^L(\mathbf{y}, j) \leq u^L(\mathbf{x}, j)$.

    iv. If any $\ell \in \widehat{S}$ is a best response to some $\mathbf{y} \in \Delta^{m-1}$, then it holds that $\tilde{u}^F(\mathbf{y}, \ell) = \max_{\ell' \in \widehat{S}} \tilde{u}^F(\mathbf{y}, \ell')$, which implies that

$$u^L(\mathbf{y}, \ell) = \min_{\ell' \in \widehat{S}} u^L(\mathbf{y}, \ell'). \tag{6}$$

Therefore, if the following also holds for the $\mathbf{y}$ in (iv), $\min_{\ell' \in \widehat{S}} u^L(\mathbf{y}, \ell') \leq u^L(\mathbf{x}, j)$, then by (6) we will have $u^L(\mathbf{y}, \ell) \leq u^L(\mathbf{x}, j)$ for every $\ell \in \widetilde{BR}(\mathbf{y}) \cap \widehat{S}$. This, together with (ii) and (iii), will imply that $u^L(\mathbf{x}, j) \geq u^L(\mathbf{y}, \ell)$ for every $\ell \in \widetilde{BR}(\mathbf{y})$. Therefore, $(\mathbf{x}, j)$ will indeed form an SSE given that $j \in \widetilde{BR}(\mathbf{x})$ by (i). We state this observation as the following lemma.

**Lemma 4.2.** *If $\min_{\ell' \in \widehat{S}} u^L(\mathbf{y}, \ell') \leq u^L(\mathbf{x}, j)$ holds for all $\mathbf{y} \in \Delta^{m-1}$ such that $\widetilde{BR}(\mathbf{y}) \cap \widehat{S} \neq \varnothing$, then the payoff matrix defined by (5) induces $(\mathbf{x}, j)$.*

The proof of Theorem 4.1 is then completed by showing the following result, the proof of which uses Farkas' Lemma [9], in particular.

**Proposition 4.3.** *If $u^L(\mathbf{x}, j) \geq M$ and $U_j(x) \neq \varnothing$, then we can construct $\widehat{S} \subseteq [n] \setminus \{j\}$ and $\alpha > 0$ in polynomial time, with which the condition of Lemma 4.2 holds for $\tilde{u}^F$ as defined in (5).*

## 5   Robustness with Respect to Equilibrium Selection

As discussed in Section 2, a weakness of BR- and payoff-inducible strategy profiles is that the resulting games may have multiple SSEs, in which case the follower depends on the leader to choose

$$u^L = \begin{pmatrix} 1/4 & 1 \\ 0 & 1/3 \\ 1/4 & 2/3 \end{pmatrix} \qquad u^F = \begin{pmatrix} 1/2 & 0 \\ 1/2 & 0 \\ 1 & 0 \end{pmatrix}$$

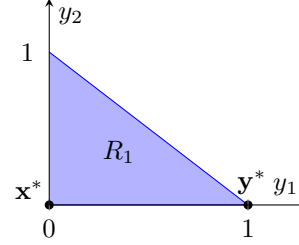

Figure 2: A game where the optimal inducible utility is 1, but the optimal *strongly* inducible utility is 0.

the SSE that maximizes his utility. To avoid this, in this section, we turn our attention to strong inducibility (see Definition 2.5) and attempt to find a payoff matrix $\tilde{u}^F$ such that $\widetilde{\mathcal{G}}$ has a unique SSE.

We begin with an example showcasing that, in general, the best strongly inducible profile can be much worse than the best payoff-inducible profile.

**Example 5.1.** Consider a $3 \times 2$ game $\mathcal{G} = (u^L, u^F)$ with the payoff matrices given in Figure 2. Note that the follower obtains positive utility only by playing his strategy 1. Now, observe that the SSE $(\mathbf{x}^*, 1)$, $\mathbf{x}^* = (0, 0, 1) \in \Delta^2$, is payoff-inducible and yields a utility of 1 for the follower: it can be induced by any payoff matrix in which strategy 1 of the follower strictly dominates all other strategies. However, such a payoff matrix will also induce other SSEs, e.g., $(\mathbf{y}^*, 1)$ with $\mathbf{y}^* = (1, 0, 0) \in \Delta^2$. Indeed, it holds that no profile of the form $(\mathbf{y}, 1)$ can be *strongly* induced, and thus the optimal utility the follower can obtain at a strongly inducible profile is 0. To see this, first note that, as seen above, if the follower claims that strategy 1 is his unique best response for all points in $\Delta^2$, then the SSE is not unique. On the other hand, if strategy 2 is a best response at some point $\mathbf{z} \in \Delta^2$, then $(\mathbf{y}, 1)$ will not be an SSE, since for the leader $u^L(\mathbf{y}, 1) < u^L(\mathbf{z}, 2)$ for any $\mathbf{y}, \mathbf{z} \in \Delta^2$. $\qquad \square$

The problem in Example 5.1 stems from the following observation: if the follower reports a payoff matrix such that strategy 1 is the unique best response for all points in the domain, then there are multiple SSEs. This can be thought of as a "degenerate" case, since it would occur with probability 0, if the payoffs of the leader were drawn uniformly at random in $[0, 1]$. We formalize this as follows.

**Definition 5.2.** A leader payoff matrix $u^L$ is said to be *max-degenerate*, if there exists $j \in [n]$ such that $|\operatorname{argmax}_{i \in [m]} u^L(i, j)| > 1$.

We next provide an example showing that even when $u^L$ is *not* max-degenerate, we cannot hope to *exactly* achieve the optimal inducible utility via a strongly inducible profile.

**Example 5.3.** Consider a $3 \times 2$ game with the leader and follower payoff matrices given in Figure 3. It is easy to check that $u^L$ is not max-degenerate. Now, observe that the maximin utility of the leader is $M = 1/2$ and is achieved at the point $\mathbf{y}^* = (\frac{1}{2}, \frac{1}{2}, 0) \in \Delta^2$. Let $\mathbf{x}^* = (0, 0, 1) \in \Delta^2$. Since $u^L(\mathbf{x}^*, 1) = 1/2 \geq M$, it follows that $(\mathbf{x}^*, 1)$ is payoff-inducible by Theorem 4.1. Indeed, the partition $(R_1, R_2)$ of $\Delta^2$ in Figure 3 shows how $(\mathbf{x}^*, 1)$ can be induced. Note that $u^F(\mathbf{x}^*, 1) = 1$, while any profile different from $(\mathbf{x}^*, 1)$ yields utility strictly less than 1 for the follower. We will now show that $(\mathbf{x}^*, 1)$ cannot be strongly induced, which implies that any strongly inducible profile gives utility strictly less than 1 to the follower. Indeed, suppose that $(\mathbf{x}^*, 1)$ is induced by some $\tilde{u}^F$. If by $\tilde{u}^F$ strategy 1 is a best response to $\mathbf{y}^*$, then $(\mathbf{x}^*, 1)$ cannot be the unique SSE, since $u^L(\mathbf{x}^*, 1) = u^L(\mathbf{y}^*, 1)$. On the other hand, if strategy 2 is the only best response to $\mathbf{y}^*$, then there exists some sufficiently small $\delta > 0$ such that strategy 2 is also a best response to $\mathbf{w}^* = (\frac{1}{2} - \delta, \frac{1}{2} + \delta, 0)$ (see Figure 3). However, this means that $(\mathbf{x}^*, 1)$ cannot be an SSE, since $u^L(\mathbf{x}^*, 1) = 1/2$ and $u^L(\mathbf{w}^*, 2) = 1/2 + \delta$. $\qquad \square$

As a result, unlike in the previous section, here we cannot hope to solve the problem exactly. However, the next theorem shows that we can approximate the optimal utility with arbitrarily good precision.

**Theorem 5.4.** *If $u^L$ is not max-degenerate, then for any $\varepsilon > 0$, the follower can strongly induce a profile $(\mathbf{x}, j)$ that yields the optimal inducible utility up to an additive loss of at most $\varepsilon$. Furthermore, a matrix $\tilde{u}^F$ strongly inducing $(\mathbf{x}, j)$ can be constructed in time polynomial in $\log(1/\varepsilon)$ (and the size of the representation of the game).*

$$u^L = \begin{pmatrix} 1 & 0 \\ 0 & 1 \\ 1/2 & 1/4 \end{pmatrix} \qquad u^F = \begin{pmatrix} 1/4 & 1/3 \\ 1/4 & 1/2 \\ 1 & 1/6 \end{pmatrix}$$

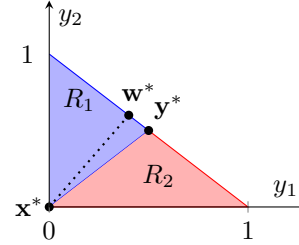

Figure 3: A non-max-degenerate game for which the optimal inducible utility cannot be achieved by any strongly inducible profile.

## 6  Conclusion and Future Work

The approach we demonstrated for a follower to optimally deceive the leader demonstrates a connection with the leader's maximin utility. From this interesting connection it is perhaps also easy to see the main idea behind the approach — to exploit information asymmetry in the game. Indeed, the degree of exploitation can be maximum: the follower can induce any SSE as long as it provides the leader the maximin utility, which is the utility a rational player can guarantee in any case. This indicates a risk of learning to make a commitment in a Stackelberg game, especially in the face of information asymmetry; in the worst case, the deceptive behavior of the follower may essentially void the learning attempt, resulting in the leader to obtain no useful information to improve her utility.

There are several directions for future work. An interesting first question that emerges from our results is how to design countermeasures to mitigate the potential loss of a learning leader, caused by possible deceptive behavior of the follower. This was considered in [20], where as a solution it was proposed that the leader could commit to a policy, which is a strategy conditioned on the report of the follower, instead of a strategy. However, in contrast to [20], where the follower's report is limited to a finite set of payoff matrices, computing the optimal policy in our model seems to be a very challenging problem. In addition, it would be nice to explore whether the optimal follower payoff matrix (or a good approximation of it) can still be computed efficiently, when additional constraints on how much he can deviate from his true payoff matrix are imposed. Finally, another interesting direction would be to perform empirical analyses to study the *average* utility gain of the follower, as well as the average loss of the leader, using both synthetic and real world data.

## Broader Impact Statement

The nature of our work is mainly theoretical, and aims to abstractly model many interesting and practical applications. Our paper can be considered as part of an emerging literature which aims to advance our understanding on the behavior of learning algorithms in the presence of strategic noise, an assumption that is arguably very realistic and appears in practice.

We are able to show that the follower can always benefit by deceiving the algorithm used by the leader to learn an equilibrium in Stackelberg games. This result has a greater impact by showing that in real-world settings which can be modeled as Stackelberg games, such as firm competition and allocating defensive resources, the design of learning algorithms needs to take into account possible ways of manipulation by the users.

## Acknowledgments and Disclosure of Funding

Georgios Birmpas is supported by the ERC Starting grant number 639945 (ACCORD), the ERC Advanced Grant 788893 AMDROMA "Algorithmic and Mechanism Design Research in Online Markets", and the MIUR PRIN project ALGADIMAR "Algorithms, Games, and Digital Markets". Jiarui Gan is supported by the EPSRC International Doctoral Scholars Grant EP/N509711/1. Alexandros Hollender is supported by an EPSRC doctoral studentship (Reference 1892947).

## Footnotes

[1]This standard assumption is justified by the fact that such tie-breaking behavior can often be enforced by an infinitesimal perturbation in the leader's strategy [40].

[2]We remark that learning the optimal strategy based on *best-response queries* may require exponentially many follower responses in the worst case [36]. However, our results are independent of issues like this, as our results hold irrespective of the algorithm that the leader uses to learn the best responses of the follower.

[3]Note that the use of $\overline{U_j(\mathbf{x})}$, instead of the set $\{\mathbf{y} \in \Delta^{m-1} : u^L(\mathbf{y}, j) \geq u^L(\mathbf{x}, j)\}$, is important: when $u^L(\mathbf{y}, j) = u^L(\mathbf{x}, j)$ for all $\mathbf{y} \in \Delta^{m-1}$, these two sets define different behaviors.

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
