[Supplementary Material]

## A  Formal Definition of Polytopal Best Response Correspondence

**Definition A.1** (Polytopal best response correspondence [22]). A best response correspondence $\widetilde{BR} : \Delta^{m-1} \to 2^{[n]} \setminus \{\varnothing\}$ is *polytopal* if it also satisfies the following:

- $\widetilde{BR}^{-1}(\ell)$ is a closed convex polytope for each $\ell \in [n]$, and

- For each $k \neq \ell$, either $\mathrm{relint}(\widetilde{BR}^{-1}(k)) \cap \mathrm{relint}(\widetilde{BR}^{-1}(\ell)) = \varnothing$ or $\widetilde{BR}^{-1}(k) = \widetilde{BR}^{-1}(\ell)$, where $\mathrm{relint}(H)$ denotes the relative interior of a set $H$.

## B  Omitted Proofs

### B.1  Proof of Lemma 3.1

*Proof.* Recall that we want to show that $u^L(\mathbf{x}, j) \geq M$ if and only if

$$u^L(\mathbf{x}, j) \geq \max_{\mathbf{y} \in \overline{U_j(\mathbf{x})}} \min_{\ell \in [n] \setminus \{j\}} u^L(\mathbf{y}, \ell) \tag{7}$$

where $M$ is the maximin utility of the leader.

We show that (7) does not hold if and only if $u^L(\mathbf{x}, j) < M$. Suppose that (7) does not hold. Then $u^L(\mathbf{x}, j) < \max_{\mathbf{y} \in \overline{U_j(\mathbf{x})}} \min_{\ell \in [n] \setminus \{j\}} u^L(\mathbf{y}, \ell)$ by definition, which implies that $U_j(\mathbf{x}) \neq \varnothing$. By the continuity of $\min_{\ell \in [n] \setminus \{j\}} u^L(\cdot, \ell)$, there exists $\mathbf{y}^* \in U_j(\mathbf{x})$ such that

$$u^L(\mathbf{x}, j) < \min_{\ell \in [n] \setminus \{j\}} u^L(\mathbf{y}^*, \ell).$$

By the definition of $U_j(\mathbf{x})$, we also have $u^L(\mathbf{x}, j) < u^L(\mathbf{y}^*, j)$. Thus,

$$u^L(\mathbf{x}, j) < \min_{\ell \in [n]} u^L(\mathbf{y}^*, \ell) \leq \max_{\mathbf{y} \in \Delta^{m-1}} \min_{\ell \in [n]} u^L(\mathbf{y}, \ell) = M.$$

Conversely, suppose that $u^L(\mathbf{x}, j) < M$. Let $\mathbf{y}^* \in \arg\max_{\mathbf{y} \in \Delta^{m-1}} \min_{\ell \in [n]} u^L(\mathbf{y}, \ell)$. Thus, $M = \min_{\ell \in [n]} u^L(\mathbf{y}^*, \ell)$, and we have

$$u^L(\mathbf{x}, j) < M = \min_{\ell \in [n]} u^L(\mathbf{y}^*, \ell) \leq u^L(\mathbf{y}^*, j)$$

which implies that $\mathbf{y}^* \in U_j(\mathbf{x})$. It follows that $M = \max_{\mathbf{y} \in \overline{U_j(\mathbf{x})}} \min_{\ell \in [n]} u^L(\mathbf{y}, \ell)$ and thus

$$u^L(\mathbf{x}, j) < \max_{\mathbf{y} \in \overline{U_j(\mathbf{x})}} \min_{\ell \in [n]} u^L(\mathbf{y}, \ell) \leq \max_{\mathbf{y} \in \overline{U_j(\mathbf{x})}} \min_{\ell \in [n] \setminus \{j\}} u^L(\mathbf{y}, \ell),$$

so (7) does not hold. $\qquad\square$

### B.2  Proof of Proposition 4.3

The proof relies on the following useful lemma.

**Lemma B.1** (Farkas' Lemma [9]). *Let $\mathbf{A} \in \mathbb{R}^{n_1 \times n_2}$ and $\mathbf{b} \in \mathbb{R}^{n_1}$. Then exactly one of the following statements is true:*

1. *there exists $\mathbf{z} \in \mathbb{R}^{n_2}$ such that $\mathbf{A}\mathbf{z} = \mathbf{b}$ and $\mathbf{z} \geq 0$;*

2. *there exists $\mathbf{z} \in \mathbb{R}^{n_1}$ such that $\mathbf{A}^\top \mathbf{z} \geq 0$ and $\mathbf{b} \cdot \mathbf{z} < 0$.*

*Proof of Proposition 4.3.* Consider any strategy profile $(\mathbf{x}, j)$ with $u^L(\mathbf{x}, j) \geq M$ and $U_j(x) \neq \varnothing$. We begin by taking care of a simple case, as an immediate corollary of Lemma 4.2.

**Corollary B.2.** *A matrix $\tilde{u}^F$ that induces $(\mathbf{x}, j)$ can be constructed in polynomial time if it holds that*

$$u^L(\mathbf{x}, j) \geq M_{-j} := \max_{\mathbf{y} \in \Delta^{m-1}} \min_{\ell \in [n] \setminus \{j\}} u^L(\mathbf{y}, \ell). \tag{8}$$

*Proof.* Let $\widehat{S} = [n] \setminus \{j\}$. Then, for every $\mathbf{y} \in \Delta^{m-1}$, we immediately obtain that

$$u^L(\mathbf{x}, j) \geq \max_{\mathbf{y} \in \Delta^{m-1}} \min_{\ell \in [n] \setminus \{j\}} u^L(\mathbf{y}, \ell) \geq \min_{\ell \in \widehat{S}} u^L(\mathbf{y}, \ell)$$

By Lemma 4.2, the payoff matrix defined by (5) (with, say, $\alpha = 1$) then induces $(\mathbf{x}, j)$, and can clearly be computed in polynomial time. $\qquad \square$

The more challenging case is when (8) does not hold (e.g., the case with the profile $(\mathbf{x}, 1)$ in Example 3.2). In what follows, we prove Proposition 4.3 by showing that there is still a choice of $\widehat{S}$ and $\alpha$ that leads to the condition in Lemma 4.2, even when (8) does not hold. Thus, from now on, we assume that

$$u^L(\mathbf{x}, j) < M_{-j}. \tag{9}$$

We define the following useful components. By Lemma 3.1 and the assumption that $u^L(\mathbf{x}, j) \geq M$, we know that

$$u^L(\mathbf{x}, j) \geq V \tag{10}$$

where

$$V = \max_{\mathbf{y} \in \overline{U_j(\mathbf{x})}} \min_{\ell \in [n] \setminus \{j\}} u^L(\mathbf{y}, \ell).$$

Since $\overline{U_j(\mathbf{x})} \neq \varnothing$, there exists $\mathbf{y}^* \in \overline{U_j(\mathbf{x})}$ such that

$$\min_{\ell \in [n] \setminus \{j\}} u^L(\mathbf{y}^*, \ell) = V, \tag{11}$$

which can be computed efficiently by solving an LP (i.e., maximize $\mu$, subject to $\mu \leq u^L(\mathbf{y}, \ell)$ for all $\ell \in [n] \setminus \{j\}$ and $\mathbf{y} \in \overline{U_j(\mathbf{x})}$). We then let

$$S = \{\ell \in [n] \setminus \{j\} \mid u^L(\mathbf{y}^*, \ell) = V\}.$$

Before we proceed, we prove two useful technical results.

**Lemma B.3.** $u^L(\mathbf{y}^*, j) = u^L(\mathbf{x}, j)$.

*Proof.* For the sake of contradiction, suppose that $u^L(\mathbf{y}^*, j) \neq u^L(\mathbf{x}, j)$. Since $\mathbf{y}^* \in \overline{U_j(\mathbf{x})}$, we have that $u^L(\mathbf{y}^*, j) \geq u^L(\mathbf{x}, j)$, so it must be that $u^L(\mathbf{y}^*, j) > u^L(\mathbf{x}, j)$.

The assumption (9) that $u^L(\mathbf{x}, j) < M_{-j}$ implies that there exists $\hat{\mathbf{y}} \in \Delta^{m-1}$ such that

$$\min_{\ell \in [n] \setminus \{j\}} u^L(\hat{\mathbf{y}}, \ell) > u^L(\mathbf{x}, j) \geq V,$$

where we also use (10). Now that $\min_{\ell \in [n] \setminus \{j\}} u^L(\mathbf{y}^*, \ell) = V$ by (11), by the concavity of $\min_{\ell \in [n] \setminus \{j\}} u^L(\cdot, \ell)$, it follows that $\min_{\ell \in [n] \setminus \{j\}} u^L(\mathbf{z}, \ell) > V$ for all $\mathbf{z}$ on the segment $[\hat{\mathbf{y}}, \mathbf{y}^*)$; $\mathbf{z} \in \Delta^{m-1}$ as $\Delta^{m-1}$ is convex. Now that we have $u^L(\mathbf{y}^*, j) > u^L(\mathbf{x}, j)$ under our assumption, when $\mathbf{z}$ is sufficiently close to $\mathbf{y}^*$, we can have $u^L(\mathbf{z}, j) \geq u^L(\mathbf{x}, j)$ and hence, $\mathbf{z} \in \overline{U_j(\mathbf{x})}$. This leads to the contradiction that

$$V = \max_{\mathbf{y} \in \overline{U_j(\mathbf{x})}} \min_{\ell \in [n] \setminus \{j\}} u^L(\mathbf{y}, \ell) \geq \min_{\ell \in [n] \setminus \{j\}} u^L(\mathbf{z}, \ell) > V. \qquad \square$$

**Lemma B.4.** $\min_{\ell \in S} u^L(\mathbf{y}, \ell) < V$ for all $\mathbf{y} \in U_j(\mathbf{x})$.

*Proof.* For the sake of contradiction, assume that there exists $\hat{\mathbf{y}} \in U_j(\mathbf{x})$ such that

$$\min_{\ell \in S} u^L(\hat{\mathbf{y}}, \ell) \geq V.$$

By assumption (9) that $u^L(\mathbf{x}, j) < M_{-j}$, there exists $\mathbf{z} \in \Delta^{m-1}$ such that $\min_{\ell \in [n] \setminus \{j\}} u^L(\mathbf{z}, \ell) > u^L(\mathbf{x}, j) \geq V$, which immediately yields the following given that $S \subseteq [n] \setminus \{j\}$ by definition:

$$\min_{\ell \in S} u^L(\mathbf{z}, \ell) > V.$$

By definition, $u^L(\mathbf{y}^*, \ell) = V$ for all $\ell \in S$, which also implies that $u^L(\mathbf{y}^*, \ell) > V$ for all $\ell \in [n] \setminus (\{j\} \cup S)$ (otherwise, we would have $\min_{\ell \in [n] \setminus \{j\}} u^L(\mathbf{y}^*, \ell) < V$). Thus, we have

$$\min_{\ell \in S} u^L(\mathbf{y}^*, \ell) = V \quad \text{and} \quad \min_{\ell \in [n] \setminus (\{j\} \cup S)} u^L(\mathbf{y}^*, \ell) > V.$$

Now consider a point $\mathbf{w}$ on the segment $(\mathbf{y}^*, \hat{\mathbf{y}}]$. Since $\mathbf{y}^* \in \overline{U_j(\mathbf{x})}$ and $\hat{\mathbf{y}} \in U_j(\mathbf{x})$, i.e., $u^L(\mathbf{y}^*, j) \geq u^L(\mathbf{x}, j)$ and $u^L(\hat{\mathbf{y}}, j) > u^L(\mathbf{x}, j)$, we have $u^L(\mathbf{w}, j) > u^L(\mathbf{x}, j)$ and hence, $\mathbf{w} \in U_j(\mathbf{x})$. In addition, by continuity, when $\mathbf{w}$ is sufficiently close to $\mathbf{y}^*$, we have

$$\min_{\ell \in [n] \setminus (\{j\} \cup S)} u^L(\mathbf{w}, \ell) > V. \tag{12}$$

By concavity of the function $\min_{\ell \in S} u^L(\cdot, \ell)$, since $\min_{\ell \in S} u^L(\mathbf{y}, \ell) \geq V$ for both $\mathbf{y} \in \{\mathbf{y}^*, \hat{\mathbf{y}}\}$, we have

$$\min_{\ell \in S} u^L(\mathbf{w}, \ell) \geq V. \tag{13}$$

Analogously, we can find a point $\mathbf{w}' \in U_j(\mathbf{x})$ on the segment $(\mathbf{w}, \mathbf{z}]$, such that (12) and (13) hold for $\mathbf{w}'$ while (13) is strict, in particular. Thus, we have

$$\min_{\ell \in [n] \setminus \{j\}} u^L(\mathbf{w}', \ell) > V = \max_{\mathbf{y} \in \overline{U_j(\mathbf{x})}} \min_{\ell \in [n] \setminus \{j\}} u^L(\mathbf{y}, \ell),$$

which is a contradiction as $\mathbf{w}' \in U_j(\mathbf{x})$. $\qquad \square$

In what follows, we use the coordinates $(y_1, \ldots, y_{m-1})$ for every point $\mathbf{y} \in \Delta^{m-1}$, i.e., we have

$$\Delta^{m-1} = \left\{ (y_1, \ldots, y_{m-1}) \in \mathbb{R}_{\geq 0} : \sum_{i=1}^{m-1} y_i \leq 1 \right\}.$$

Accordingly, we can write the utility function as

$$u^L(\mathbf{y}, \ell) = \mathbf{g}_\ell \cdot \mathbf{y} + u^L(m, \ell),$$

where $\mathbf{g}_\ell \in \mathbb{R}^{m-1}$ and its $i$-th component is $g_{\ell,i} = u^L(i, \ell) - u^L(m, \ell)$; "$\cdot$" denotes the inner product. Hence, we have

$$u^L(\mathbf{y}, \ell) = \mathbf{g}_\ell \cdot (\mathbf{y} - \mathbf{y}^*) + u^L(\mathbf{y}^*, \ell) = \begin{cases} \mathbf{g}_\ell \cdot (\mathbf{y} - \mathbf{y}^*) + V & \text{if } \ell \in S \\ \mathbf{g}_j \cdot (\mathbf{y} - \mathbf{y}^*) + u^L(\mathbf{x}, j) & \text{if } \ell = j \end{cases} \tag{14}$$

where $u^L(\mathbf{y}^*, \ell) = V$ for all $\ell \in S$ by the definition of $S$, and $u^L(\mathbf{y}^*, j) = u^L(\mathbf{x}, j)$ by Lemma B.3. Note that since $U_j(\mathbf{x}) \neq \varnothing$, it must be that $\mathbf{g}_j \neq 0$.

We also write the $m$ boundary conditions that define $\Delta^{m-1}$ as $\mathbf{e}_i \cdot \mathbf{y} \geq \beta_i$. Namely, for each $i \in [m-1]$, let $\mathbf{e}_i \in \mathbb{R}^{m-1}$ be the $i$-th unit vector and $\beta_i = 0$, while $\mathbf{e}_m = (-1, \ldots, -1) \in \mathbb{R}^{m-1}$ and $\beta_m = -1$. Thus, $\Delta^{m-1} = \{\mathbf{y} \in \mathbb{R}^{m-1} : \mathbf{e}_i \cdot \mathbf{y} \geq \beta_i \text{ for } i \in [m]\}$. Let

$$B = \{i \in [m] : \mathbf{e}_i \cdot \mathbf{y}^* = \beta_i\}$$

be the set of boundary conditions that are tight for $\mathbf{y}^*$. Note that for any $\mathbf{y} \in \Delta^{m-1}$ we have

$$\mathbf{e}_i \cdot (\mathbf{y} - \mathbf{y}^*) \geq 0 \quad \text{for all } i \in B. \tag{15}$$

We can now prove the following result using Farkas' Lemma (Lemma B.1), which allows us to express $-\mathbf{g}_j$ as a non-negative linear combination of $\mathbf{g}_\ell$'s and $\mathbf{e}_i$'s.

**Lemma B.5.** $-\mathbf{g}_j$ *can be expressed as a non-negative linear combination of* $\{\mathbf{g}_\ell : \ell \in S\} \cup \{\mathbf{e}_i : i \in B\}$, *i.e.* $-\mathbf{g}_j = \sum_{\ell \in S} \lambda_\ell \mathbf{g}_\ell + \sum_{i \in B} \mu_i \mathbf{e}_i$, *where* $\lambda_\ell \geq 0$ *and* $\mu_i \geq 0$.

*Proof.* We use Farkas' Lemma (Lemma B.1) and let $n_1 = m - 1$ and $n_2 = |S| + |B|$. The columns of $\mathbf{A}$ are exactly the vectors $\{\mathbf{g}_\ell : \ell \in S\} \cup \{\mathbf{e}_i : i \in B\}$. We set $\mathbf{b} = -\mathbf{g}_j$. Note that the first alternative of Farkas' Lemma immediately yields the statement we want to prove. Thus, we set out to prove that the second alternative cannot hold.

Assume, for the sake of contradiction, that there exists $\mathbf{z} \in \mathbb{R}^{m-1}$ such that $\mathbf{A}^\top \mathbf{z} \geq 0$ and $\mathbf{b} \cdot \mathbf{z} < 0$, i.e., $\mathbf{g}_\ell \cdot \mathbf{z} \geq 0$ for all $\ell \in S$, $\mathbf{e}_i \cdot \mathbf{z} \geq 0$ for all $i \in B$, and $\mathbf{g}_j \cdot \mathbf{z} > 0$.

Then, by picking $\delta > 0$ sufficiently small, it holds for $\mathbf{y} = \mathbf{y}^* + \delta\mathbf{z}$ that:

- By (14), we have the following for all $\ell \in S$:

$$u^L(\mathbf{y}, \ell) = \mathbf{g}_\ell \cdot (\mathbf{y} - \mathbf{y}^*) + V = \delta \mathbf{g}_\ell \cdot \mathbf{z} + V \geq V.$$

In addition,

$$u^L(\mathbf{y}, j) = \mathbf{g}_j \cdot (\mathbf{y} - \mathbf{y}^*) + u^L(\mathbf{y}^*, j) = \delta \mathbf{g}_j \cdot \mathbf{z} + u^L(\mathbf{x}, j) > u^L(\mathbf{x}, j).$$

- $\mathbf{y} \in \Delta^{m-1}$: For $i \in B$, we immediately obtain that $\mathbf{e}_i \cdot \mathbf{y} = \mathbf{e}_i \cdot (\mathbf{y}^* + \delta \mathbf{z}) \geq \mathbf{e}_i \cdot \mathbf{y}^* = \beta_i$, which means that these boundary conditions are satisfied. For $i \in [m] \setminus B$, we know that $\mathbf{e}_i \cdot \mathbf{y}^* > \beta_i$ and thus by picking $\delta > 0$ small enough, we can ensure that $\mathbf{e}_i \cdot \mathbf{y} = \mathbf{e}_i \cdot \mathbf{y}^* + \delta(\mathbf{e}_i \cdot \mathbf{z}) \geq \beta_i$.

Thus, it follows that $\mathbf{y} \in U_j(\mathbf{x})$ and $\min_{\ell \in S} u^L(\mathbf{y}, \ell) \geq V$. But this cannot hold according to Lemma B.4. $\qquad \square$

We can now complete the proof of Proposition 4.3.

We first express $-\mathbf{g}_j$ as a non-negative linear combination of the vectors $\{\mathbf{g}_\ell : \ell \in S\} \cup \{\mathbf{e}_i : i \in B\}$. By Lemma B.5 we know that this is possible and it is easy to see that we can find the coefficients in polynomial time (e.g. by solving an LP). We thus obtain $-\mathbf{g}_j = \sum_{\ell \in S} \lambda_\ell \mathbf{g}_\ell + \sum_{i \in B} \mu_i \mathbf{e}_i$, where $\lambda_\ell \geq 0$ for every $\ell \in S$ and $\mu_i \geq 0$ for every $i \in B$. Let $\widehat{S} = \{\ell \in S : \lambda_\ell > 0\}$. We will argue that $\widehat{S} \neq \varnothing$.

Observe that since now $-\mathbf{g}_j = \sum_{\ell \in S} \lambda_\ell \mathbf{g}_\ell + \sum_{i \in B} \mu_i \mathbf{e}_i$ and, by (15), we have $\mathbf{e}_i \cdot (\mathbf{y} - \mathbf{y}^*) \geq 0$ for all $\mathbf{y} \in \Delta^{m-1}$ and $i \in B$, it follows that, for all $\mathbf{y} \in \Delta^{m-1}$, we have

$$
\begin{aligned}
-\mathbf{g}_j \cdot (\mathbf{y} - \mathbf{y}^*) &= \sum_{\ell \in S} \lambda_\ell \mathbf{g}_\ell \cdot (\mathbf{y} - \mathbf{y}^*) + \sum_{i \in B} \mu_i \mathbf{e}_i \cdot (\mathbf{y} - \mathbf{y}^*) \\
&\geq \sum_{\ell \in S} \lambda_\ell \mathbf{g}_\ell \cdot (\mathbf{y} - \mathbf{y}^*) \\
&= \sum_{\ell \in \widehat{S}} \lambda_\ell \mathbf{g}_\ell \cdot (\mathbf{y} - \mathbf{y}^*), \qquad\qquad\qquad (16)
\end{aligned}
$$

where the last transition is due to the fact that $\lambda_\ell = 0$ for all $\ell \in S \setminus \widehat{S}$, as implied by the definition of $\widehat{S}$.

Since $U_j(\mathbf{x}) \neq \varnothing$, consider any $\mathbf{y} \in U_j(\mathbf{x})$. By definition, this means that $u^L(\mathbf{y}, j) > u^L(\mathbf{x}, j)$, which further implies that $\mathbf{g}_j \cdot (\mathbf{y} - \mathbf{y}^*) > 0$ since $u^L(\mathbf{y}, j) = \mathbf{g}_j \cdot (\mathbf{y} - \mathbf{y}^*) + u^L(\mathbf{x}, j)$ by (14). By (16), we then have

$$\sum_{\ell \in \widehat{S}} \lambda_\ell \mathbf{g}_\ell \cdot (\mathbf{y} - \mathbf{y}^*) < 0.$$

Hence, $\widehat{S} \neq \varnothing$.

It remains to show that with the above $\widehat{S}$ and, in particular, $\alpha = 1/\lambda_k$ (recall that $k \in \arg\min_{\ell \in \widehat{S}} u^L(\mathbf{x}, \ell)$), the condition in Lemma 4.2 holds, i.e., we prove that $\min_{\ell \in \widehat{S}} u^L(\mathbf{y}, \ell) \leq u^L(\mathbf{x}, j)$ holds for all $\mathbf{y} \in \Delta^{m-1}$ such that $\widetilde{BR}(\mathbf{y}) \cap \widehat{S} \neq \varnothing$.

For the sake of contradiction, suppose that there exists $\mathbf{y} \in \Delta^{m-1}$ such that $\widetilde{BR}(\mathbf{y}) \cap \widehat{S} \neq \varnothing$, but $u^L(\mathbf{y}, \ell) > u^L(\mathbf{x}, j)$ for all $\ell \in \widehat{S}$. By (10), we have $u^L(\mathbf{x}, j) \geq V$, and thus $u^L(\mathbf{y}, \ell) > V$ for all $\ell \in \widehat{S}$. By (14), we have $u^L(\mathbf{y}, \ell) = \mathbf{g}_\ell \cdot (\mathbf{y} - \mathbf{y}^*) + V$; thus, $\mathbf{g}_\ell \cdot (\mathbf{y} - \mathbf{y}^*) > 0$ for all $\ell \in \widehat{S}$.

Using (16) and the fact that $k \in \widehat{S}$ by our choice, we then obtain

$$-\mathbf{g}_j \cdot (\mathbf{y} - \mathbf{y}^*) \geq \sum_{\ell \in \widehat{S}} \lambda_\ell \mathbf{g}_\ell \cdot (\mathbf{y} - \mathbf{y}^*) \geq \lambda_k \mathbf{g}_k \cdot (\mathbf{y} - \mathbf{y}^*).$$

By (14), we have

$$u^L(\mathbf{x}, j) - u^L(\mathbf{y}, j) = -\mathbf{g}_j \cdot (\mathbf{y} - \mathbf{y}^*).$$

Recall that it is defined that $\tilde{u}^F(\mathbf{y}, j) = -u^L(\mathbf{y}, k) + \alpha \left( u^L(\mathbf{x}, j) - u^L(\mathbf{y}, j) \right)$ as in (5). Using the above two equations and (14), we then obtain the following:

$$\begin{aligned}
\tilde{u}^F(\mathbf{y}, j) &= -u^L(\mathbf{y}, k) + \alpha \left( u^L(\mathbf{x}, j) - u^L(\mathbf{y}, j) \right) \\
&= -\mathbf{g}_k \cdot (\mathbf{y} - \mathbf{y}^*) - V - \alpha \mathbf{g}_j \cdot (\mathbf{y} - \mathbf{y}^*) \\
&\geq -V + (\alpha \lambda_k - 1)\mathbf{g}_k \cdot (\mathbf{y} - \mathbf{y}^*) \\
&= -V.
\end{aligned}$$

However, by (5) we also have $\tilde{u}^F(\mathbf{y}, \ell) = -u^L(\mathbf{y}, \ell)$ if $\ell \in \widehat{S}$, which implies that for all $\ell \in \widehat{S}$ it holds that

$$\tilde{u}^F(\mathbf{y}, j) \geq -V > -u^L(\mathbf{y}, \ell) = \tilde{u}^F(\mathbf{y}, \ell).$$

Hence, $\widetilde{BR}(\mathbf{y}) \cap \widehat{S} = \varnothing$, which contradicts our assumption. $\qquad\square$

## B.3   Proof of Theorem 5.4

*Proof of Theorem 5.4.* Let $(\mathbf{x}^*, j)$ be a payoff-inducible profile that yields the optimal inducible payoff for the follower. By Theorem 4.1, such a profile can be computed in polynomial time.

We begin by solving the following LP.

$$\begin{aligned}
\max_{\delta, \mathbf{x}} \quad & \delta \\
\text{subject to} \quad & \mathbf{x} \in \Delta^{m-1} \\
& u^F(\mathbf{x}, j) \geq u^F(\mathbf{x}^*, j) - \varepsilon \\
& u^L(\mathbf{x}, j) = u^L(\mathbf{x}^*, j) + \delta
\end{aligned} \tag{17}$$

Note that this LP can be solved in time polynomial in $\log(1/\varepsilon)$. Furthermore, note that the polytope of feasible points is not empty since $\delta = 0$ and $\mathbf{x} = \mathbf{x}^*$ satisfy all the constraints. Finally, the LP is not unbounded since $\delta$ can be at most $\max_{i \in [m]} u^L(i, j) - u^L(\mathbf{x}^*, j)$.

In the rest of this proof let $\delta$ and $\mathbf{x}$ denote an optimal solution to this LP. Note that we can in particular assume that $\mathbf{x}$ is a vertex of the convex polytope $P_\delta = \{\mathbf{y} \in \Delta^{m-1} : u^L(\mathbf{y}, j) = u^L(\mathbf{x}^*, j) + \delta\}$. Indeed, given a solution $\delta, \mathbf{x}$ to LP (17), if $\mathbf{x}$ is not a vertex of $P_\delta$, then we consider the LP

$$\begin{aligned}
\max_{\mathbf{y}} \quad & u^F(\mathbf{y}, j) \\
\text{subject to} \quad & \mathbf{y} \in \Delta^{m-1} \\
& u^L(\mathbf{y}, j) = u^L(\mathbf{x}^*, j) + \delta
\end{aligned}$$

It is known that a solution of an LP that is also a vertex of the feasible polytope can be computed in polynomial time. Note that in this case the feasible polytope is exactly $P_\delta$. Let $\mathbf{y}$ be an optimal solution that is a vertex of $P_\delta$. We know that $\mathbf{x} \in P_\delta$ and $u^F(\mathbf{x}, j) \geq u^F(\mathbf{x}^*, j) - \varepsilon$, which implies that $u^F(\mathbf{y}, j) \geq u^F(\mathbf{x}^*, j) - \varepsilon$. But this means that $\delta, \mathbf{y}$ is also an optimal solution to the original LP (17). Thus, by letting $\mathbf{x} := \mathbf{y}$, we indeed have that $\mathbf{x}$ is a vertex of the convex polytope $P_\delta$.

Let us first handle the case where $\delta = 0$ by showing that $(\mathbf{x}^*, j)$ itself can be strongly induced. Since $\delta = 0$, it follows that $U_j(\mathbf{x}^*) = \varnothing$. Indeed, if there exists $\hat{\mathbf{y}} \in \Delta^{m-1}$ with $u^L(\hat{\mathbf{y}}, j) > u^L(\mathbf{x}^*, j)$, then there exists $\mathbf{y}$ on the segment $(\mathbf{x}^*, \hat{\mathbf{y}}]$ such that $u^F(\mathbf{y}, j) \geq u^F(\mathbf{x}^*, j) - \varepsilon$ (when $\mathbf{y}$ is sufficiently close to $\mathbf{x}^*$) and $u^L(\mathbf{y}, j) > u^L(\mathbf{x}^*, j)$, a contradiction to the optimality of $\delta = 0$. Now, given that $U_j(\mathbf{x}^*) = \varnothing$, we have that $u^L(\mathbf{y}, j) \leq u^L(\mathbf{x}^*, j)$ for all $\mathbf{y} \in \Delta^{m-1}$. But since $u^L$ is not max-degenerate (in the sense of Definition 5.2), it follows that in fact $u^L(\mathbf{y}, j) < u^L(\mathbf{x}^*, j)$ for all $\mathbf{y} \in \Delta^{m-1} \setminus \{\mathbf{x}^*\}$. Thus, if the follower always best responds with strategy $j$, then $(\mathbf{x}^*, j)$ will be the unique SSE. As seen before, it is easy to implement this behavior by reporting $\tilde{u}^F(i, j) = 1$ and $\tilde{u}^F(i, \ell) = 0$ for all $i \in [m]$ and $\ell \in [n] \setminus \{j\}$.

In the rest of this proof, we consider the case $\delta > 0$ and show that $(\mathbf{x}, j)$ can be strongly induced. Since $u^F(\mathbf{x}, j) \geq u^F(\mathbf{x}^*, j) - \varepsilon$, this means that at $(\mathbf{x}, j)$ the follower achieves the optimal inducible utility up to an additive error of $\varepsilon$. Using the same notation as in the proof of Proposition 4.3, we let

$$B = \{i \in [m] : \mathbf{e}_i \cdot \mathbf{x} = \beta_i\}$$

denote the set of boundary conditions of $\Delta^{m-1}$ that are tight for $\mathbf{x}$. Note that since $\mathbf{x}$ is a vertex of the polytope $P_\delta$, it follows that $B \neq \varnothing$. We let $\mathbf{h} = \sum_{i \in B} \mathbf{e}_i$. As in the proof of Proposition 4.3, we have that for all $\mathbf{y} \in \Delta^{m-1}$ it holds that

$$\mathbf{h} \cdot (\mathbf{y} - \mathbf{x}) = \sum_{i \in B} \mathbf{e}_i \cdot (\mathbf{y} - \mathbf{x}) \geq 0. \tag{18}$$

Furthermore, since $\mathbf{x}$ is a vertex of $P_\delta$, it follows that for all $\mathbf{y} \in P_\delta \setminus \{\mathbf{x}\}$ there exists $i \in B$ such that $\mathbf{e}_i \cdot (\mathbf{y} - \mathbf{x}) > 0$, and thus

$$\mathbf{h} \cdot (\mathbf{y} - \mathbf{x}) > 0. \tag{19}$$

Indeed, if $\mathbf{e}_i \cdot (\mathbf{y} - \mathbf{x}) = 0$ for all $i \in B$ for some $\mathbf{y} \in P_\delta \setminus \{\mathbf{x}\}$, this would contradict the fact that $\mathbf{x}$ is a vertex of $P_\delta$ (i.e. the unique point in $P_\delta$ for which the boundary conditions in $B$ are tight).

We are now ready to construct the payoff matrix reported by the follower. Pick $k \in \operatorname{argmin}_{\ell \in [n] \setminus \{j\}} u^L(\mathbf{x}, \ell)$ arbitrarily. For all $\mathbf{y} \in \Delta^{m-1}$ let

$$\tilde{u}^F(\mathbf{y}, \ell) = \begin{cases} -u^L(\mathbf{y}, \ell) & \text{if } \ell \in [n] \setminus \{j\} \\ -u^L(\mathbf{y}, k) + \alpha \left( u^L(\mathbf{x}, j) - u^L(\mathbf{y}, j) \right) - \mathbf{h} \cdot (\mathbf{y} - \mathbf{x}) & \text{if } \ell = j \end{cases} \tag{20}$$

where $\alpha = \left( 2 \max_{i \in [m]} \max_{\ell \in [n]} \left| u^L(i, \ell) \right| + m \right) / \delta > 0$. Note that we can compute the payoff matrix corresponding to this utility function in polynomial time. In the remainder of this proof, we show that $(\mathbf{x}, j)$ is the unique SSE of the game $\left( u^L, \tilde{u}^F \right)$.

Clearly, $j$ is a best response at $\mathbf{x}$, since

$$\tilde{u}^F(\mathbf{x}, j) = -u^L(\mathbf{x}, k) = -\min_{\ell \in [n] \setminus \{j\}} u^L(\mathbf{x}, \ell) = \max_{\ell \in [n] \setminus \{j\}} \tilde{u}^F(\mathbf{x}, \ell),$$

by the choice of $k$.

Next, let us show that if $j$ is a best response at some $\mathbf{y} \in \Delta^{m-1} \setminus \{\mathbf{x}\}$, then $u^L(\mathbf{y}, j) < u^L(\mathbf{x}, j)$. Indeed, if $j$ is a best response at $\mathbf{y}$, then in particular $\tilde{u}^F(\mathbf{y}, j) \geq \tilde{u}^F(\mathbf{y}, k)$, which implies that

$$\alpha \left( u^L(\mathbf{x}, j) - u^L(\mathbf{y}, j) \right) \geq \mathbf{h} \cdot (\mathbf{y} - \mathbf{x}). \tag{21}$$

Since $\mathbf{h} \cdot (\mathbf{y} - \mathbf{x}) \geq 0$ by (18), and $\alpha > 0$, it follows that $u^L(\mathbf{x}, j) \geq u^L(\mathbf{y}, j)$. It remains to show that $u^L(\mathbf{x}, j) \neq u^L(\mathbf{y}, j)$. But if $u^L(\mathbf{x}, j) = u^L(\mathbf{y}, j)$, then $\mathbf{y} \in P_\delta \setminus \{\mathbf{x}\}$ and so by (19) we have $\mathbf{h} \cdot (\mathbf{y} - \mathbf{x}) > 0$, which contradicts (21).

Finally, it remains to show that if $\ell \in [n] \setminus \{j\}$ is a best response at some $\mathbf{y} \in \Delta^{m-1}$, then it must be that $u^L(\mathbf{y}, \ell) < u^L(\mathbf{x}, j)$: Indeed, if $\ell \in [n] \setminus \{j\}$ is a best response at $\mathbf{y}$, then in particular $\tilde{u}^F(\mathbf{y}, j) \leq \tilde{u}^F(\mathbf{y}, \ell)$, which by (20) means that

$$\begin{aligned} \alpha \left( u^L(\mathbf{x}, j) - u^L(\mathbf{y}, j) \right) &\leq -u^L(\mathbf{y}, \ell) + u^L(\mathbf{y}, k) + \mathbf{h} \cdot (\mathbf{y} - \mathbf{x}) \\ &\leq -u^L(\mathbf{y}, \ell) + u^L(\mathbf{y}, k) + \|\mathbf{h}\|_2 \|\mathbf{y} - \mathbf{x}\|_2 \\ &\leq 2 \max_{i \in [m]} \max_{\ell' \in [n]} |u^L(i, \ell')| + \sqrt{m - 1} \sqrt{m - 1} \\ &\leq \alpha \delta \end{aligned}$$

by the choice of $\alpha$. Thus, we obtain that $u^L(\mathbf{x}, j) - u^L(\mathbf{y}, j) \leq \delta$, which implies that $u^L(\mathbf{y}, j) \geq u^L(\mathbf{x}^*, j)$, i.e. $\mathbf{y} \in \overline{U_j(\mathbf{x}^*)}$ (since $U_j(\mathbf{x}^*) \neq \varnothing$). Since $(\mathbf{x}^*, j)$ is payoff-inducible, which means that $u^L(\mathbf{x}^*, j) \geq M$, we can use Lemma 3.1 to obtain

$$u^L(\mathbf{x}, j) = u^L(\mathbf{x}^*, j) + \delta > u^L(\mathbf{x}^*, j) \geq \min_{\ell' \in [n] \setminus \{j\}} u^L(\mathbf{y}, \ell') = u^L(\mathbf{y}, \ell)$$

where the last equality comes from the fact that $\ell$ is a best response at $\mathbf{y}$, i.e., in particular $\tilde{u}^F(\mathbf{y}, \ell) = \max_{\ell' \in [n] \setminus \{j\}} \tilde{u}^F(\mathbf{y}, \ell')$. $\qquad \square$