[Reviews · NeurIPS 2020]

Review 1

Summary and Contributions: This paper studies the optimal way for the follower to deceive the leader/learner in a Stackelberg game. The learner cannot observe the follower's payoff matrix so that she can only learn it from interacting with the learner's best response to her strategy. As a result, the follower can deceive the learner by acting as if he has a different payoff matrix. The authors examine three levels of inducibility. For each level, the authors provide efficient algorithms to compute the optimal way to deceive the learner.

Strengths: (1) This paper is well-written, clear, and easy to follow. The reviewer appreciates the nicely constructed examples and figures. (2) The authors provide full characterizations and efficient algorithms for all three levels of inducibility. (3) This paper concerns the problem of deceiving a learning algorithm, which is relevant and interesting to the NeurIPS community. [After response: thank you for the response.]

Weaknesses: .

Correctness: The reviewer did not check the proofs in the appendix, but all the results seem sound and plausible in hindsight. The nicely constructed examples also provide evidence that the results are correct.

Clarity: This paper is generally well written, clear, and easy to follow.

Relation to Prior Work: The authors do a good job of surveying the literature.

Reproducibility: Yes

Additional Feedback:


Review 2

Summary and Contributions: In Stackelberg games, a follower can employ adversarial deception to manipulate the best response calculated by the leader. Past work has explored this problem under a finite payoff space; this paper explores the general case of this problem where the payoff space is infinite. Specifically, the follower optimally deceives a learning leader by misreporting his payoff matrix. The authors assume that the leader has full knowledge of her own payoff matrix and is trying to learn the payoff matrix of the follower. The authors establish that the follower may play a deceptive strategy to his own benefit in this general case, thus improving his utility. They define a set of inducible strategy profiles, which the follower can deceive the leader into learning, and then maximize his own utility from the set of inducible strategy profiles. They look at both payoff-inducible and best response-inducible strategies. In best response inducibility, for example, the leader sets her strategy by querying the best responses of the follower.

Strengths: The authors show that optimal deception is possible in a more generalizable case of Stackelberg games, in which the payoff space is infinite. Additionally, they show that the follower can perform optimal deception by constructing a payoff matrix which induces a unique SSE, driving the leader to follow a single strategy. They consider best-response inducibility, and payoff-inducibility. The proofs are sound and thoroughly built up through clear definitions and explanations along the way. This work sits at a growing intersection of machine learning and game theory.

Weaknesses: It would be useful for the authors to highlight any key insights in their theoretical analysis. There is no experimental section; this paper would benefit greatly from experimental results that demonstrate increased performance of the adversary by acting deceptively. For example, [19, 20, 34] which the authors claim to build upon all include empirical evaluations. This would help to show the utility gained when moving to an infinite payoff space (compared to [20]). Would like discussion of implications of results at the end instead of only briefly listing future work.

Correctness: Yes, the claims are convincing and the proofs appear sound.

Clarity: The paper is well-written with clear notation. Small suggestions: -- unnecessary comma on line 313

Relation to Prior Work: Yes, the authors point to two recent papers [19, 34] that look at the general version of optimal deception without constraints on the payoff matrix. However, those papers focus on resource allocation problems (Stackelberg security games), not Stackelberg games more broadly. In other work [20] with a polynomially finite payoff matrix, the follower can simply enumerate all possible rewards in polynomial time to determine an optimal payoff.

Reproducibility: Yes

Additional Feedback: Regarding broader impact, the authors should acknowledge that although their work is theoretical, Stackelberg games have been deployed in real-world settings such ones they mention of competition in a firm and allocating defensive resources. Thus these strategies could, theoretically, be employed by real-world adversaries.


Review 3

Summary and Contributions: The paper considers the problem of optimally deceiving a learning leader in a general stackelberg game. A two player stackelberg game is played between a leader and a follower, where the leader first commits a (mix) strategy and the leader plays with a best response. This paper follows a continuous line of work on assuming the leader does not have prior knowledge of the payoff matrix of the follower and can only make a best response query, i.e., commit an arbitrary mix strategy and receive a (fake) best response. The paper studies the problem from the follower side, i.e., they try to tackle with how the follower can optimally deceiving a leader using a learning algorithm, and achieves the best possible utility for its own. The paper proposes several constraints on the type of actions a follower can take, i.e., BR-inducibility, payoff-inducibility, strong-inducibility, from strong to weak. They develop a complete picture on the power of the follow. For all of the above indelibility, the paper develops algorithm optimal algorithm for the follower.

Strengths: The paper is theoretically sound, and some of the results are interesting.

Weaknesses: The model is somehow not quite natural, and needs more explanation. For example, a learning leader for stackelberg game usually takes exponential best response sample, what is the point of making exponential fake best response and deceiving the leader just for one single shot game? I suggest the author provides more concrete explanation on why this model is practical and useful.

Correctness: The proof seems correct, though I do not check them thoroughly.

Clarity: Generally yes, but some definition and theorem needs more explanation

Relation to Prior Work: I suggest the author puts more words on explaining previous work on sample complexity of Stackelberg game or Stackelberg security game instead of putting some general reference on query complexity of Nash equilibrium (they are less relevant to this paper).

Reproducibility: Yes

Additional Feedback: As I suggest above, the author should consider (1) give more explanation on the game theoretical model and address why they are useful in practice, (2) provide more explanations on some of the definition and theorem statement, (3) rewrite the related work section and put more words on sample complexity of Stackelberg game or Stackelberg security game. ------------------------------ I increase my score and vote for weak accept. Meanwhile, I suggest the author provide more concrete examples on the practicality of the model and more explanation some previous work if the paper get accepted.


Review 4

Summary and Contributions: There has been an important line of work on learning mixed-strategy commitment in Stackelberg games (particularly, security games) using best response oracles. Recently, there has been a flurry of papers considering the problem of optimal deception, when the adversary aims to cause the leader to play suboptimally by faking best responses, although in restricted settings. This paper presents the first general treatment of the problem, showing that optimal deception can be done in polynomial time in arbitrary Stackelberg games, and that one can even induce unique SSE, albeit now only approximately optimally.

Strengths: + This is a significant and non-trivial generalization of prior art in terms of optimal deception of Stackelberg leader learning how to act. + The technical contribution settles a major open problem in the area. + The paper also presents a nice comprehensive picture of the problem, with results considering equilibrium selection issue.

Weaknesses: The main weakness of the work is the lack of motivation for considering BR_p. Of course, if you read the paper sequentially, this consideration makes sense. However, the paper ultimately shows that the optimal deception problem can be solved using an LP that is identical whether one considers BR, BR_p, or payoff inducibility. Thus, the discussion of BR_p doesn't really add anything of substance to the paper; it would have sufficed to deal with BR and payoff inducibility (indeed, the equivalence of these seems to be understood in prior literature; however, I haven't seen a crisp proof of this before).

Correctness: The claims appear correct.

Clarity: The paper is mostly well written, the caveat about BR_p discussion excepting (it should just be removed from the paper).

Relation to Prior Work: The paper clearly differs from prior work.

Reproducibility: Yes

Additional Feedback: The main question is: what is the purpose of the section of polytopal BR inducibility? It seems that it doesn't add much value to the discussion, given that everything is ultimately equivalent. Post rebuttal: thank you for your effort; I still believe it's a strong paper.

[Author Response · NeurIPS 2020]

We wish to thank all of the reviewers for their insightful comments and suggestions. Please find below our detailed responses to different comments made by each of the reviewers.

**Reviewer 1**

We thank the reviewer for the positive assessment of our paper!

**Reviewer 2**

We thank the reviewer for their helpful comments about highlighting the key insights of our analysis and implications of our results. We will revise our discussion accordingly both in the introduction and in the conclusion of the paper. In short, our approach shows that there is a strong connection between the follower's ability to optimally deceive the leader and the leader's maximin utility. In particular, by exploiting the information asymmetry in the game, the follower can induce any SSE as long as the leader is guaranteed to obtain his maximin utility. This shows that there is an inherently high risk in learning to commit optimally in a Stackelberg game. In the worst case, the deceptive behavior of the follower may essentially void the learning attempt, which means that the leader has no useful information to improve his utility.

Regarding potential experimental analysis: Indeed, experiments would be a good way to showcase the utility gain of the follower. Please observe that Example 2.2 already shows a rather simple instance where the utility gain of the follower, as well as the utility loss of the leader, can be arbitrarily large. Nevertheless, empirical analysis is definitely an interesting direction to study the *average* utility gain of the follower, as well as the average loss of the leader, using both synthetic and real world data. The utility loss of the leader is also a very interesting theoretical question, which we have mentioned in the conclusion. Considering the density of the paper and its focus on the already non-trivial task of finding the optimal deceiving strategy of the follower, we believe that these are excellent questions for future work.

We also thank the reviewer for the suggestion about the broader impact statement. We will make the requested adjustments!

**Reviewer 3**

The reviewer is right that learning the optimal strategy based on *best-response queries* may require exponentially many follower responses in the worst case; this result is due to Peng et al. [36]. However, please note that this is not always the case. In particular, Peng et al. have showed that in certain circumstances, such as when the number of actions $n$ or $m$ is small, or when the smallest feasible region is small (as in the work of Letchford et al. [30]), only a polynomial number of queries is required by the learning algorithms. The key assumption adopted in this paper as well as a series of other previous papers (i.e., [3, 6, 30, 39]) is that the cost of learning can be safely ignored because the game is expected to be repeated sufficiently many times; hence, the learned optimal strategy is not used just in one single-shot game.

Most importantly, our results also apply to learning based on *payoff queries*, which can be done very efficiently. In fact, a remarkable aspect of our results is that the problem of optimally deceiving the leader to maximize the utility of the follower turns out to be easily solvable, *no matter* what learning algorithm, or what type of queries, is used by the leader. We believe that the problem we study is also a fundamental theoretical question in Stackelberg games, i.e., how a follower can optimally disguise his payoffs in the presence of information asymmetry, which is interesting in its own right.

We will extend the discussion of the related work to include cases where learning based on various types of queries can be done efficiently. We will also add a more crisp motivation about our modeling assumptions, as well as intuitions about our techniques and the implications of our results (as was also requested by reviewer 2).

**Reviewer 4**

It is true that the discussion of polytopal BR inducibility could easily have been omitted from the paper. We decided to include it in an attempt to ease the transition from best-response inducibility to payoff inducibility, and introduce useful geometric intuition for our more involved results. In a certain sense, it serves as a "warm up" for the subsequent sections that are more technical. In our revisions we will make this progression and intuition more clear.

[Meta-Review · NeurIPS 2020]

Four knowledgeable referees agree that the paper should be accepted (though one referee, R3, remains borderline). R3 continues to argue that the amount of time required to learn to deceive the learner can be exponential. While the authors argue that sometimes it would not be, the reviewer's criticism is valid, though this does not invalidate the contribution the paper makes. Thus, I concur that the paper should be accepted. The paper is theoretical in nature, expanding on a wide range of results. I believe the paper could be presented effectively as a poster.